# Exploring user and stakeholder perspectives from South Africa and Uganda to refine microarray patch development for HIV PrEP delivery and as a multipurpose prevention technology

**Ayesha Ismail**[1], **Sarah Magni**[1], **Anne Katahoire**[2], **Florence Ayebare**[2], **Godfrey Siu**[2], **Fred Semitala**[3], **Peter Kyambadde**[4], **Barbara Friedland**[5], **Courtney Jarrahian**[6], **Maggie Kilbourne-Brook**[6]*

1 Genesis Analytics, Johannesburg, South Africa, 2 Child Health and Development Centre, Makerere University, Kampala, Uganda, 3 Ministry of Health, Kampala, Uganda, 4 Department of Internal Medicine, College of Health Sciences, Makerere University, Kampala, Uganda, 5 Population Council, New York, New York, United States of America, 6 PATH, Seattle, Washington, United States of America

* mkilbou@path.org

## Abstract

### Background

Oral HIV pre-exposure prophylaxis (PrEP) is highly effective, but alternative delivery options are needed to reach more users. Microarray patches (MAPs), a novel drug-delivery system containing micron-scale projections or "microneedles" that deliver drugs via skin, are being developed to deliver long-acting HIV PrEP and as a multipurpose prevention technology to protect from HIV and unintended pregnancy. We explored whether MAP technology could meet user and health system needs in two African countries.

### Methods

Researchers in South Africa and Uganda conducted 27 focus group discussions, 76 mock-use exercises, and 31 key informant interviews to explore perceptions about MAPs and specific features such as MAP size, duration of protection, delivery indicator, and health system fit. Participants included young women and men from key populations and vulnerable groups at high risk of HIV and/or unintended pregnancy, including adolescent girls and young women; female sex workers and men who have sex with these women; and men who have sex with men. In Uganda, researchers also recruited young women and men from universities and the community as vulnerable groups. Key stakeholders included health care providers, sexual and reproductive health experts, policymakers, and youth activists. Qualitative data were transcribed, translated, coded, and analyzed to explore perspectives and preferences about MAPs. Survey responses after mock-use in Uganda were tabulated to assess satisfaction with MAP features and highlight areas for additional refinement.

**Funding:** 1 of 1 Funder: -Initial of author who received the award: CJ -Grant number: AID-OAA-A-17-00015. -Funder name: US Agency for International Development (USAID) -Website: https://www.usaid.gov/ NO. The funder had no role in study design, data collection and analysis, decision to publish, or preparation of the manuscript.

**Competing interests:** The authors have declared that no competing interests exist.

## Results

All groups expressed interest in MAP technology, reporting perceived advantages over other methods. Most participants preferred the smallest MAP size for ease of use and discreetness. Some would accept a larger MAP if it provided longer protection. Most preferred a protection duration of 1 to 3 months or longer; others preferred 1-week protection. Upper arm and thigh were the most preferred application sites. Up to 30 minutes of wear time was considered acceptable; some wanted longer to ensure the drug was fully delivered. Self-administration was valued by all groups; most preferred initial training by a provider.

## Conclusions

Potential users and stakeholders showed strong interest in/acceptance of MAP technology, and their feedback identified key improvements for MAP design. If a MAP containing a high-potency antiretroviral or a MAP containing both an antiretroviral and hormonal contraceptive is developed, these products could improve acceptability/uptake of protection options in sub-Saharan Africa.

## Introduction

Forty years into the epidemic, HIV/AIDS continues to critically affect the health and welfare of individuals, families, and communities. Although advances in HIV testing and treatment have reduced mortality and improved health outcomes, additional progress is needed for HIV prevention options to avert new infections among at-risk individuals.

Sub-Saharan Africa continues to be disproportionately affected by HIV, with women and girls representing more than 60% of new infections in 2020 [1]. Although new infections in eastern and southern Africa have dropped nearly 40% since 2010 [2], young women continue to be at high risk. In sub-Saharan Africa, six in seven newly HIV-infected adolescents (15 to 19 years) are girls [1]. Girls and young women aged 15 to 24 years are twice as likely to be living with HIV as young men [1]. Multiple contributing factors include unequal gender norms and power dynamics, stigma associated with accessing sexual and reproductive health (SRH) services, limited access to education and financial resources, which influences sexual decision-making, and the complexities of life for young persons, which take precedence over HIV prevention [2]. Female sex workers (FSW), men who have sex with men (MSM), and male partners of sex workers also are populations at increased risk of HIV due to high-risk behaviors [3–7].

Antiretroviral (ARV) drugs for oral pre-exposure prophylaxis (PrEP) reduce the incidence of HIV infections when taken consistently [8]. Although the World Health Organization has recommended oral PrEP since 2015 for persons at significant risk of HIV [9], the daily oral pill regimen is challenging for some, and inconsistent adherence leads to reduced effectiveness [10,11]. Additional prevention options have achieved regulatory approvals and are moving closer to introduction in sub-Saharan African countries. The one-month dapivirine vaginal ring, which provides discreet, long-acting protection for women, is currently being rolled out in sub-Saharan Africa [12] based on the results of two trials demonstrating 30% reduced HIV risk [13,14]. Injectable cabotegravir has been shown to be even more effective for HIV prevention than oral PrEP [15,16] and provides long-acting protection for all genders. Other prevention options in development include PrEP implants, vaginal inserts, and vaginal film [17].

While consumer choice for HIV prevention products is increasing, product developers are continuing to advance additional delivery systems for HIV PrEP to better meet the needs of diverse populations ranging from FSW and MSM to adolescent girls and young women (AGYW)—the latter being least able to use existing methods [18].

While there is clearly a need for additional HIV prevention options, there is also strong rationale for developing prevention products that address multiple health indications at the same time (i.e., multipurpose prevention technologies (MPTs) [19–21]. Many sub-Saharan African countries that report a high incidence of HIV also report a high unmet need for contraception and low use (less than 20%) of modern contraceptive methods [22–25], leaving women in this region vulnerable to both unintended pregnancy and HIV. MPTs could help address the needs of women who face both risks [19,26]. The World Health Organization acknowledges the need for a more holistic approach to HIV prevention that integrates other services, such as contraception, particularly for populations at greatest risk [27]. Meanwhile, countries are moving toward more holistic service delivery [28–30]. MPTs fit well into this vision. Research suggests that women may be more interested in a product that could protect from pregnancy and HIV rather than a product that protects from HIV alone [31–36]. A recent analysis of the potential market for a dual prevention pill (i.e., PrEP combined with a contraceptive) in 15 sub-Saharan African countries estimates that this pill could result in up to a ten-fold increase in use over oral PrEP [21].

Microarray patch (MAP) technology could address the same needs. MAPs are a novel drug-delivery system being developed for prevention and/or treatment of many health indications [37–39]. MAPs deliver drug(s) through micro-projections or "microneedles" that pierce the top layer of the skin (Fig 1). Once the microarray dissolves and the drug is released into the body, the end user removes and discards the MAP backing (target wear time is less than 20 minutes). MAPs have the potential to improve acceptability and adherence by enabling painless drug delivery that is discreet and provides long-term protection. MAPs could also potentially reduce the burden on health care systems as they are designed to allow for self-administration by end users, likely after initial training by health care providers.

Preclinical studies show that the technology platform can deliver ARV drugs [40]. Studies have also provided proof of concept for delivery of a contraceptive MAP containing a progestin [41]. Developers of contraceptive MAPs have reported user preference for MAP characteristics from Nigeria and India [42,43] and acceptability and preference results for contraceptive MAP applicator designs and characteristics from users in The Gambia, Malawi, Uganda, and United Kingdom [44]. Unlike MAPs being developed for other applications—such as delivery of vaccines and highly potent drugs for migraines and influenza, which can be postage-stamp sized [45–47], and long-acting contraceptive MAPs, which are somewhat larger [48,49]—an even larger MAP size will be needed for HIV prevention assuming use of current ARVs [50]. Thus, the usability and acceptability of larger MAPs needs to be understood early in the development process. This study aimed to explore the perceptions of and preferences around MAP technology for delivery of an ARV for HIV prevention and as an MPT (Fig 2) among user groups at potential risk of HIV infection or both HIV and unintended pregnancy in South Africa and Uganda. Researchers also explored the usability of MAP prototypes among various potential user groups to identify refinements and assess whether MAPs could meet user needs for HIV prevention or as an MPT.

A growing body of literature emphasizes end-user acceptability as an important driver of product uptake, adherence, and continuation of sexual and reproductive health products [51–55]. PATH's human-centered design process aims to ensure that products in development address the diverse needs of all stakeholders who ultimately influence whether new products will be procured, supplied, and integrated into health systems in low- and middle-income countries.

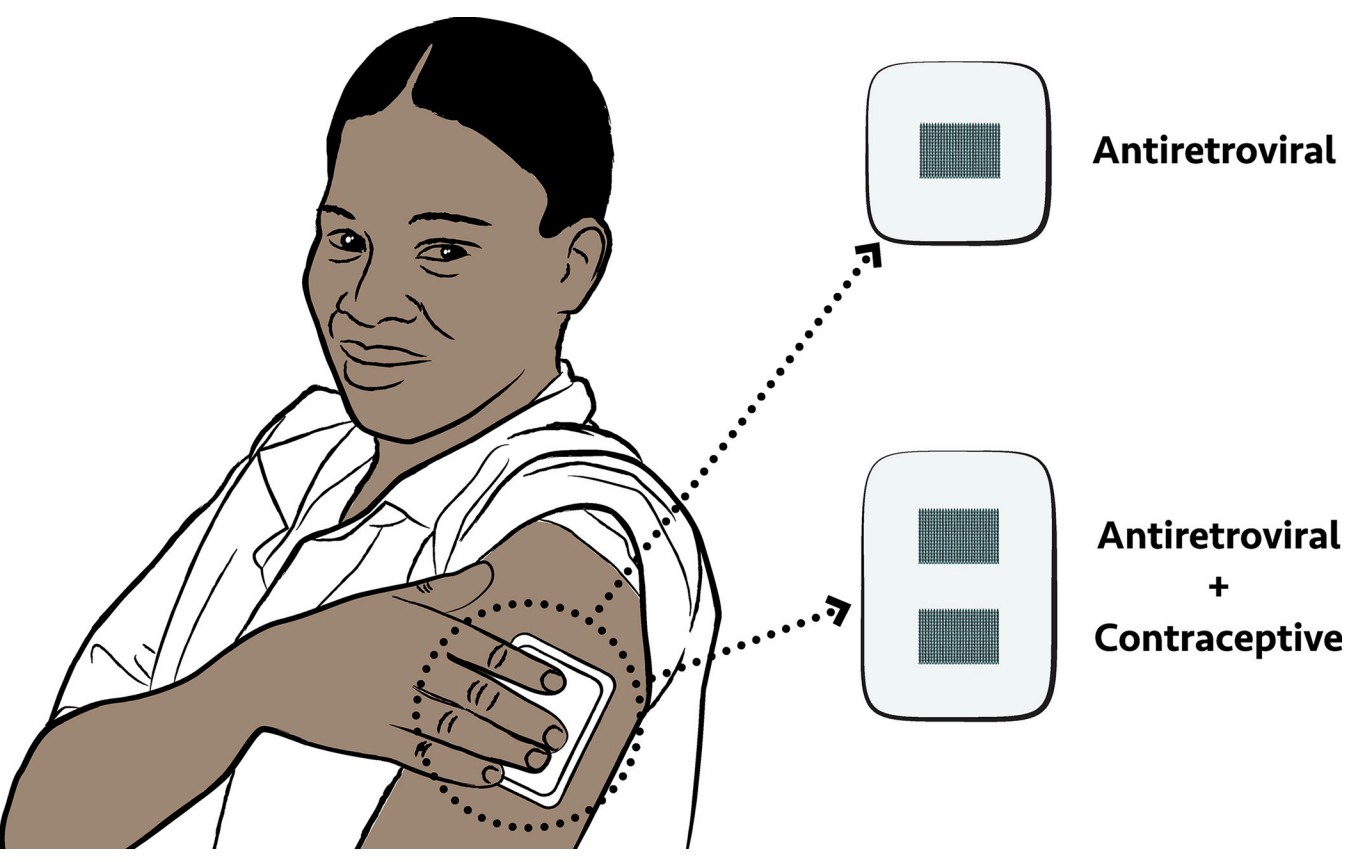

**Fig 1. Diagram of the application of a dissolving microarray patch.**

## Materials and methods

### Study design

We used cross-sectional, mixed-methods research design to explore issues pertaining to a MAP technology concept for delivery of an ARV for HIV PrEP alone or an ARV plus progestin for hormonal contraception. Specifically, our objectives were to (1) explore the hypothetical acceptability of MAP technology for HIV PrEP and as an MPT among end users (including AGYW, FSW, and MSM,) and youth activists as influencers, (2) explore the usability of MAP prototypes among end users to inform product design, and (3) assess the potential programmatic fit of MAP technology for HIV PrEP and as an MPT among policymakers, health care providers, and SRH experts.

We selected South Africa and Uganda because they have high HIV incidence, are at different stages of PrEP introduction, and represent a slightly different contraceptive method mix (with Uganda having a higher unmet need for family planning overall), reflecting a variety of user needs [56–59]. In both countries, we conducted focus group discussions (FGDs) and mock-use exercises followed by semi-structured, in-depth interviews (IDIs) with potential users; mock-use exercises followed by IDIs with health care providers; and key informant interviews (KIIs) with SRH experts, policymakers, youth activists, and health care providers (the health care providers' participation spanned each type of session except FGDs).

Because the formulation for a PrEP MAP and an MPT MAP is still in preclinical stages and product features are not finalized, we explored MAP prototype features that would most likely

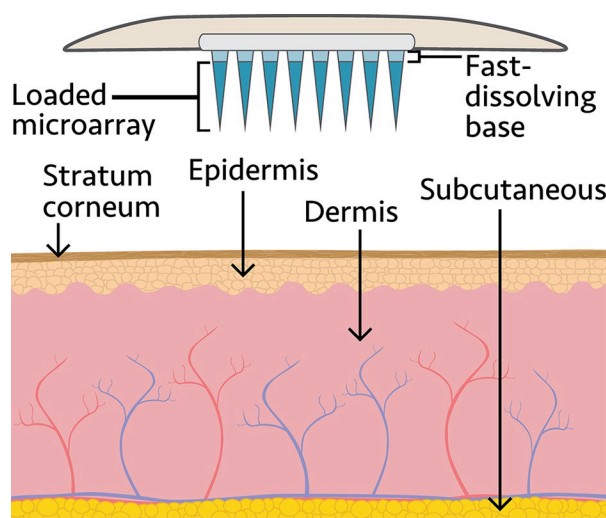

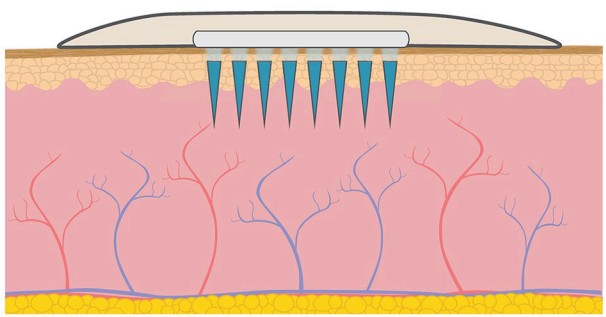

Patch is applied and microarray base quickly dissolves.

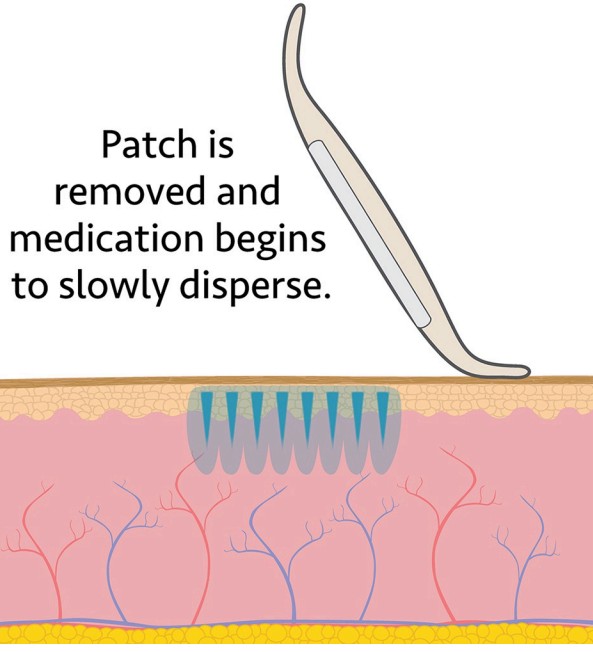

Patch is removed and medication begins to slowly disperse.

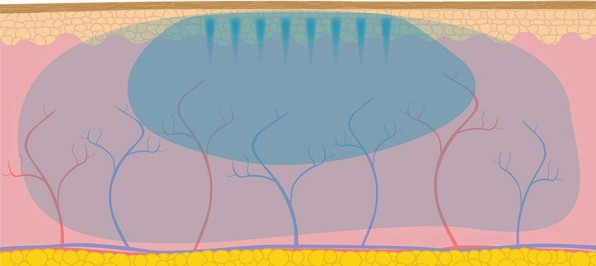

Microarray slowly dissolves, and medication is dispersed over time.

**Fig 2. Illustration of potential intradermal microarray patch delivery systems for HIV pre-exposure prophylaxis and as a multipurpose prevention technology.**

influence usability and acceptability. Through FGDs and mock use, we explored users' needs and preferences for features such as MAP size, duration of protection, wear time, site of application, delivery indicator to confirm the MAP had been applied correctly, potential for self-application, and the packaging and instructions for use. We also assessed interest in MAPs for HIV prevention relative to MAPs as an MPT to protect against HIV and unintended pregnancy.

To explore FGD participants' preferences and perceptions related to the size of the MAP, we created images of three potential MAP sizes (small 6.8 cm x 6.8 cm; medium 11 cm x 11 cm, and large 13.9 cm x 13.9 cm). The sizes were based on modeling informed by preclinical

studies of the estimated potential MAP sizes required for a protection duration of 1 week to 1 month, using the ARV cabotegravir [50,60]. The prototypes contained no drugs or microneedles.

For the mock-use exercises, which were used to evaluate usability, the prototype MAPs included a delivery indicator that was intended to provide a tactile and visual cue that the user had applied sufficient pressure to push the (in this case, hypothetical) microneedles through the skin. The prototypes represented the small size MAP dimensions. Participants in the mock-use exercises evaluated either a "spot crush" delivery indicator or a "slide crush" delivery indicator (Fig 3) and responded to questions about their experiences.

These first-generation prototypes were intended to give an example of how the delivery indicator might look, feel, and function. The goal was to explore the usability of the design, identify refinements, and assess whether MAPs could meet user needs for HIV prevention and as an MPT. Draft instructions for use, composed of images and simple text translated into local languages, were developed for the mock-use exercises.

The research teams developed IDI guides and mock-use exercise questions, which were then pretested and further refined. In Uganda, the tools were translated into Luganda; in South Africa tools were presented in English but a member of the research team spoke the local languages to facilitate understanding. The tools used in each country covered the same overall topics, but questions and procedures were adapted for the different contexts. For example, in South Africa, researchers captured user preferences primarily during the FGDs; in Uganda, participants reported satisfaction with product features after mock-use exercises using a 5-point Likert-type scale.

## Study sites and recruitment

In South Africa, assessments were implemented in three provinces: Gauteng, KwaZulu Natal, and Mpumalanga. In each province, study participants (women aged 18 to 25 years; men aged 18 to 35 years) were recruited from an urban center and either a peri-urban or rural setting. In Uganda, assessments were implemented in two districts, Kampala (urban and peri-urban) and Wakiso (rural) (participants were women aged 18 to 25 years; men aged 18 to 40 years). Table 1 summarizes the sites and settings in which the assessments took place.

In both countries, the study teams recruited participants who had existing relationships with program sites and clinics, such that staff members knew the participants came for services fitting with the study objectives. Participants were recruited using purposive sampling based on gender, age, and the participants' perception of risk for HIV infection and/or unintended pregnancy. In both countries, participants included AGYW, FSW, and men as partners to FSW. The Uganda team also included MSM and university students (men and women).

Key informants recruited from the health sector included health care providers (e.g., doctors, nurses, pharmacists, and community health workers) and SRH experts based on their involvement in the delivery of PrEP, HIV care, reproductive health care, and family planning services in private and public facilities in urban, peri-urban, and rural settings. We also recruited key informants from the Ministry of Health in Uganda (e.g., policymakers who make decisions on PrEP programming in the country) and community-based organizations in South Africa involved in advocacy for family planning, PrEP, or sexual and reproductive health and rights.

## Data collection

In South Africa, a team of two researchers conducted the mock-use exercises, FGDs, and IDIs in each province. Each field pair was composed of two South African women. Activities were

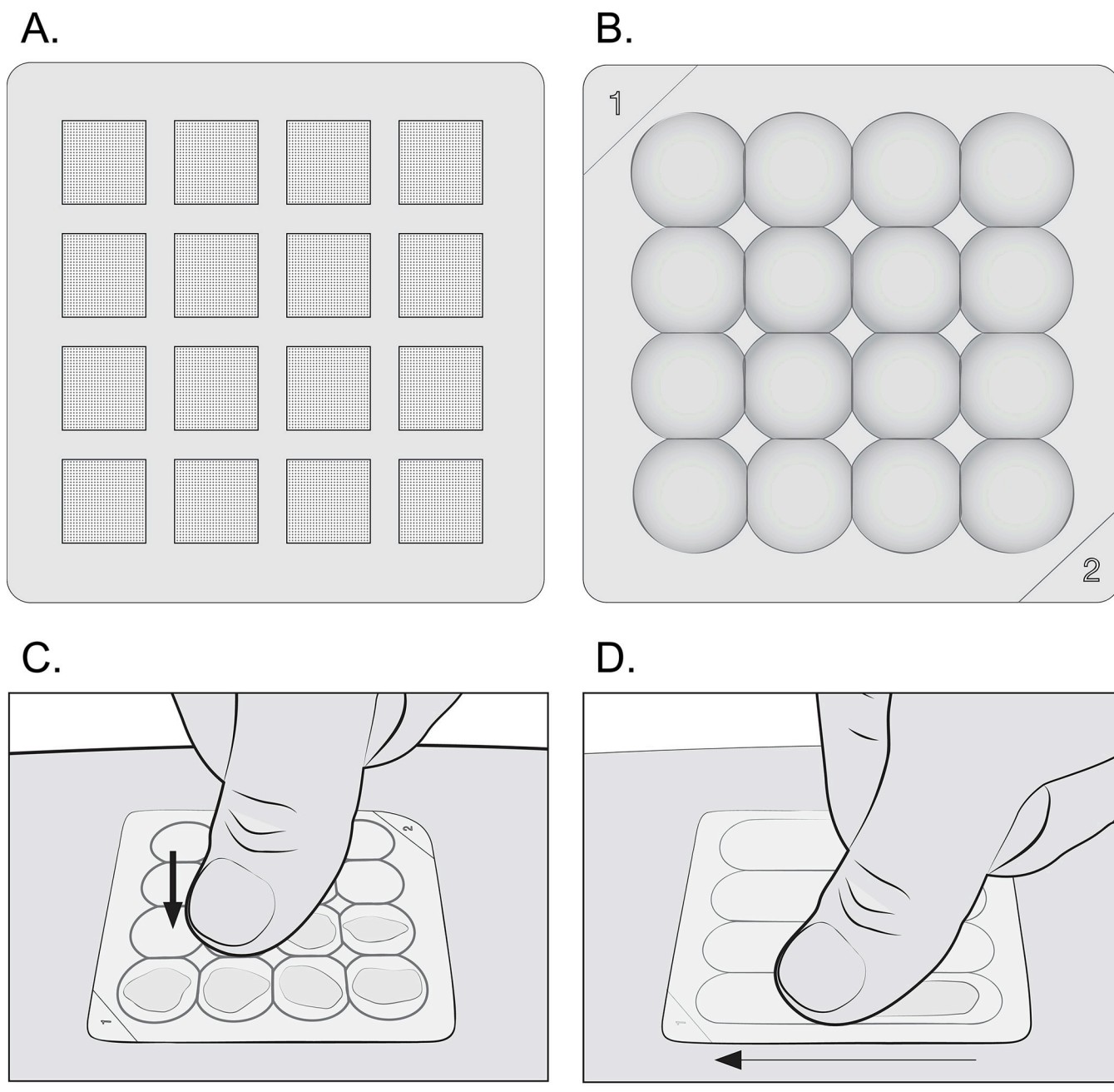

**Fig 3.** Graphical representations of the two types of microarray patch (MAP) prototypes in our study: (A) squares of simulated micro-projection arrays on the skin-facing side of the MAPs (which, for demonstration purposes, had no microneedles or drug); (B) view of the top side of the spot-crush delivery indicator design; (C) illustration from instructions for use showing how to apply the spot-crush prototype; (D) illustration from instructions for use showing how to apply the slide-crush prototype.

conducted in either English or isiZulu, depending on the preference of the participants, and were conducted at sites such as community centers, meeting rooms of HIV prevention programs and SRH organizations, or meeting spaces near facilities convenient for participants. In

**Table 1. Study sites and settings.**

| South Africa Provinces | Site | Setting |
|---|---|---|
| *KwaZulu-Natal* | City of Durban (eThekwini) | Urban |
| | Valley of 1000 Hills | Rural |
| *Gauteng* | City of Johannesburg | Urban |
| | Carletonville, West Rand | Peri-urban |
| *Mpumalanga* | City of Mbombela | Urban |
| | Gert Sibande | Peri-urban |
| **Uganda Districts** | **Site** | **Setting** |
| *Kampala* | MARPI Clinic,* Mulago National Referral Hospital | Urban |
| | Kawaala Health Centre IV | Urban |
| | Makerere | Urban |
| | Kalerwe | Peri-urban |
| *Wakiso* | Bugema | Rural |

* The MARPI Clinic (Most At-Risk Population Clinic) is an initiative of Uganda's Ministry of Health situated within Mulago National Referral Hospital. It provides HIV and SRH services to key and vulnerable populations.

Uganda, a team of four experienced researchers—two women and two men, working in mixed pairs—conducted the research. Activities were conducted either in open spaces near the clinic or university identified in collaboration with participants to ensure they felt comfortable, or at sites such as a community hall to allow privacy and freedom of discussion. In both countries, researchers conducted the national-level KIIs by telephone or in-person depending on availability. The FGDs lasted approximately 90 minutes with 5–10 participants per group in South Africa and 6–8 participants in Uganda. The FGDs were stratified by gender, age, employment, education, and risk factors, and in Uganda also by PrEP use/non-use and religion. The KII were individual interviews lasting 45–60 minutes. The mock-use observation with follow-up interview were conducted with individual participants and lasted 30–45 minutes.

During the mock-use exercises, the researchers observed participants, documented use errors (i.e., actions or lack of actions taken by the user that resulted in unintended outcomes), use difficulties (i.e., actions that were difficult or took a long time), and close calls (i.e., instances in which a use error almost occurred but did not) with the MAP, including the delivery indicator and instructions for use. After the mock-use exercises, researchers interviewed each participant to explore their experiences in applying the MAP prototypes. In Uganda, participants also completed a 5-point Likert-type scale survey to capture perceptions about prototype features after mock use. Participants in South Africa did not complete a survey of satisfaction with features use but responded to questions during interviews after mock-use exercises.

Notably, the mock-use exercise participants in South Africa had previously taken part in an FGD and seen a demonstration of how to apply the MAP, whereas, in Uganda, the mock-use exercise participants were naïve users, having neither taken part in an FGD nor seen a demonstration of how to apply the MAP.

During the KIIs, we explored perceptions that policymakers and SRH experts have about MAP technology and their views regarding MAPs as a delivery option for PrEP, MAPs as an MPT, and the potential for integrating both types of MAPs into the current health care system.

## Data management and analysis

All data from the FGDs and KIIs were audio-recorded, transcribed, and translated. The audio recordings were transferred to password-protected computers and deleted from the recorders.

All data were de-identified to protect participant identity and maintain confidentiality. In South Africa, the team developed a draft coding framework based on the first set of interviews conducted, then developed a conceptual framework and the final coding framework. Using Microsoft Excel, the research team conducted thematic content analysis. The transcripts were divided among the team for analysis, with a senior team member conducting quality assurance.

In Uganda, the team uploaded the transcripts into ATLAS.ti (version 8.5) qualitative software for coding. Using codes based on the priority themes and topics in the interview/discussion guides, the research team generated and reviewed reports to identify common patterns and emerging themes. The team then constructed matrices to compare the findings across different categories of participants. The Likert-style survey tool administered after the mock-use was analyzed by calculating response frequencies. Results were used to guide the debrief interview and highlight areas for additional product refinement.

## Ethics considerations

In South Africa, ethics approval was obtained from the University of Witwatersrand Human Research Ethics Committee (Medical) (Clearance Certificate No. M190721), and permission to access health facilities at the provincial and district level was granted through the National Health Research Database. In Uganda, ethics approval was received from the Makerere University School of Medicine Research Ethics Committee (REC REF 2019–112) and the Uganda National Council for Science and Technology (UNCST SS5090). All participants provided written informed consent prior to the FGDs, mock-use exercises, and interviews. Participants in South Africa were provided refreshments and reimbursement for travel costs.

## Results

To assess hypothetical acceptability and usability and programmatic fit, we conducted 27 FGDs (17 in South Africa, 10 in Uganda), 76 mock-use exercises and follow-up interviews/IDIs (40 in South Africa, 36 in Uganda), and 31 KIIs (18 in South Africa, 13 in Uganda) among young women and men from key populations and vulnerable groups at high risk of HIV and/or unintended pregnancy, as well as other key stakeholders. Participants were from urban/peri-urban and rural locations, and represented a range of ages, education, socio-economic and relationship status, and PrEP experience (Table 2). During FGDs and mock use, participants reported on preferences for key features that could influence use and acceptability of a future MAP product: MAP size, duration of protection; wear time; site of application, delivery indicator. They also provided feedback about the Instructions for Use (IFU) and packaging.

### Perception of MAP technology

Overall, participants from all target groups in both countries were interested in and accepting of MAP technology, even though it was a new concept and they had few examples of existing products with which to compare it. Potential users were enthusiastic about the possibility of additional HIV prevention and contraceptive options. Because MAPs have the potential to painlessly deliver long-acting protection that could be discreet and self-administered, participants believed MAPs could address the challenges of other delivery systems such as oral pills which require daily dosing, injectables, and implants which are provider-dependent methods involving some pain, and condoms which are perceived as interfering with sexual spontaneity and/or pleasure.

**Table 2. Activities completed, by country, population, and number of participants.**

| SOUTH AFRICA study population | Activity | Number of sessions implemented | Number of participants |
|---|---|---|---|
| *Adolescent girls and young women (aged 18–19 years)* | FGD | 5 | 34 |
| | MUE, IDI | 8 | 8 |
| *Adolescent girls and young women (aged 20–24 years)* | FGD | 6 | 41 |
| | MUE, IDI | 10 | 10 |
| *Female sex workers* | MUE, IDI | 4 | 4 |
| *Male partners of sex workers* | FGD | 6 | 41 |
| | MUE, IDI | 9 | 9 |
| *Health care providers* | KII | 10 | 10 |
| | MUE, IDI | 9 | 9 |
| *SRH experts and policymakers* | KII | 6 | 6 |
| *Youth activists* | KII | 2 | 2 |
| **Total South Africa** | | **75** | **174** |
| UGANDA study population | Activity | Number of sessions | Number of participants |
| *Adolescent girls and young women (community) (aged 18–25 years)* | FGD | 1 | 9 |
| | MUE, IDI | 9 | 9 |
| *Female sex workers* | FGD | 2 | 12 |
| | MUE, IDI | 9 | 9 |
| *Men who have sex with men (aged 18–40 years)* | FGD | 2 | 12 |
| | MUE, IDI | 9 | 9 |
| *Female university students (aged ≤ 20 years)* | FGD | 2 | 17 |
| *Male university students\* (aged ≤ 20 years)* | FGD | 2 | 16 |
| *Male youth (community)\* (aged 18–25 years)* | FGD | 1 | 10 |
| *Health care workers* | KII | 10 | 10 |
| | MUE, IDI | 9 | 9 |
| *Policymakers* | KII | 3 | 3 |
| **Total Uganda** | | **59** | **125** |
| **Grand totals for both countries combined** | | **FGD: 27**<br>**MUE, IDI: 76**<br>**KII: 31** | **299** |

Key: FGD, focus group discussion; IDI, in-depth interview; KII, key informant interview; MUE, mock-use exercise; SRH, sexual and reproductive health.

\* Males from university and community recruited as end-users and as partners who influence women's choice for HIV protection and contraception.

"The MAP is going to be very good for us, we can decide for ourselves to use it without relying on our boyfriends to use [a] condom."–Female university student; Uganda

"Sometimes our partners refuse to use condoms, wanting live sex and they pay more for that and because you need the money you just agree. So, this MAP will be good for us, we can protect ourselves without fearing HIV. Some of us have been using oral PrEP but those tablets are big and difficult to swallow, they smell, and they have terrible side effects."–FSW; Uganda

"Adherence is a problem to some people. . .I think this patch will be convenient as it will be easier than taking a pill every day."–Health care provider; Mpumalanga, South Africa

"It is flexible. You can put it in your bag and move with it anywhere. You can apply it by yourself, anytime and still do your work. It is timesaving and flexible."–MSM; Uganda

Health care providers, SRH experts, and policymakers in both countries felt this technology could be integrated into the health system and expand access, improve adherence, and help reduce stress on over-burdened health care providers, especially if self-administration is feasible. Sexual and reproductive health experts and policymakers highlighted that the MPT would align well with current HIV prevention policies and guidelines due to the focus on combination prevention. The most common considerations mentioned regarding integrating the MAP MPT into the health system were capacity and training of health care providers, distribution and storage of the MAPs, need for repeat HIV testing, and cost of the product. As one health care provider in Uganda mentioned, "It will help because it will be one stone hitting two birds so it will be important especially in the high-risk populations."

Other health care provider perspectives included:

"This will be very helpful to our programs, and people will be very eager to use it for prevention of HIV transmission. . .it will be a better option than the current ones where some potential clients are afraid of coming to the clinic."–Health care provider; Uganda

"I think it is a brilliant idea that will take away from the tablets because the tablets can be frustrating. You can't plan sex, so at least with the patch you can instantly think I'm still covered for the next month or so. . .the longer duration the better. People generally go for the longer family planning options."–Health care provider; KwaZulu-Natal, South Africa

"It can be delivered through all clinics that provide family planning and STI [sexually transmitted infection] services if it is at no cost. If it is for sale, you will not get the targets. . .for those of lower socioeconomic status they won't buy them. Honestly even a condom they can't go and buy."–Health care provider; Uganda

## Perceived benefits/concerns

End users perceived the MAP as easy to use, discreet, and potentially painless. The concept of long-acting protection was appealing since this would reduce clinic visits, decrease or eliminate the burden of daily oral pills, and improve adherence.

"People will like it because it is easy and you don't have to take something else [food]; you just put it on, wait, and remove."–AGYW; South Africa

"It is easy to use. . .you don't have to use it every day like tablets. . .but the MAP will stay in the body. . .until you get another one."–MSM; Uganda

"I believe it is easy to use and another thing is that it won't be a burden to clients because swallowing a tablet daily, the size is a little bigger, it is bothersome to take a tablet, sometimes you can easily forget but this one can be easily used and is not painful."–Health care provider; Uganda

Nearly all categories of respondents liked the possibility of self-administration and noted that individuals could apply the MAP in the privacy of their home without having to explain themselves to anyone. Most respondents wanted to learn how to use the MAP first from a health care provider but would feel comfortable with self-application after that. Some health care providers, especially in Uganda, felt that some users would not feel comfortable applying the MAP themselves, even after initial training, and would want to come to a clinic.

"We would like to self-administer. First time the nurse must show us, then after that we can do it ourselves. It will help if it's in clinics, pharmacies, and hospitals."–AGYW; South Africa

"What I like about this MAP is that once you master the instructions you can use it even without the help of a health worker. You can use it at your own time."–MSM; Uganda

"I would prefer to do this by myself. It's simple, it's easy. With the instructions, I would be able to do this."–FSW; South Africa

Participants also had questions and raised some concerns. Since these prototypes did not contain microneedles and did not deliver drug, some participants were confused about how the MAP would work. Some wanted proof that the MAP would fully deliver the drug as expected and they were not sure whether they could trust the MAP. They wanted assurance that it would be safe and effective and that there would be no pain or long-term scarring from the MAP. They wondered if MAPs would have fewer side effects than other drug-delivery methods.

"We like this MAP or patch and the way you have explained how it works. But many of us have never heard of it or seen anyone using it. We do not know how well it works and whether we can trust it. For example, we know that even if PrEP tablets are difficult to swallow and have side effects, we know that they work and that once you take them you know that you are protected. Will this MAP be as effective? How much medication will be in each MAP? We will need that reassurance before using it."–MSM; Uganda

"The drugs will accumulate in our bodies. What I know, it also has its own side effect. It may only show small scratches or lines on the surface of the skin, but the scratches can become severe if you overuse the MAP."–FSW; South Africa

## User needs/preferences

Participants shared their perspectives about needs and preferences regarding MAP features, including MAP size, duration of protection; wear time; site of application, delivery indicator, and packaging. They also evaluated the IFU.

**MAP size.** Across all FGD user groups, most participants preferred the smallest prototype due to ease of use, especially for self-application and discreetness—though for some participants, especially MSM in Uganda and some AGYW in South Africa, the smallest size MAP prototype was already considered too large to be discreet.

"It needs to be small because some girls don't want to show that they are using this."–AGYW; Mpumalanga, South Africa

"The problem with the larger one is that it is too big and there are too many dots to crush."–AGYW; Mpumalanga, South Africa

When AGYW in South Africa were asked if they had to choose between two small-sized MAPs or one large MAP, they preferred a large MAP. Participants in the FGDs felt numerous MAPs would increase the potential for error, including removing one before sufficient time had passed. In Uganda, even among participants who expressed some willingness to use a larger-size MAP (i.e., MSM and AGYW), they said the larger size made it intimidating, unattractive, likely to draw unnecessary attention and stigma, difficult to carry around and conceal while worn, and that it limited the application site options and would be difficult to apply independently. Those who preferred the larger-size MAP assumed it would carry larger quantities of drug and would likely provide a longer duration of protection. Longer protection meant reduced frequency of re-application, fewer refills, less frequent visits to health clinics, savings on time and transport, and extra protection for unplanned sexual encounters.

**Duration of protection.** Across both countries, most participants in the FGDs and mock-use exercises desired a 1-month minimum duration of protection, while many preferred 2 to 3 months or longer. Participants who desired longer protection cited ease of use and fewer clinic visits, which would save time and transport costs. Among female university students and AGYW who preferred longer protection, they said this would ensure they were less stressed.

"Two months would be better because even with the injection it's 2 months or 3 months. It will also help avoid the inconvenience of going to the facility every month."–AGYW; Mpumalanga, South Africa

"Some of us stay in the homelands so the clinics are far, so we won't have to keep going to the clinic."– Male partner; KwaZulu-Natal, South Africa

"Why can't we have a small product with lots of drugs in it to last longer?"–AGYW; KwaZulu Natal, South Africa

For FGD participants in South Africa, duration of protection was considered the most important feature of MAPs. In Uganda, FGD participants' preferences varied somewhat across different groups, but most still preferred at least 1 month of protection. Yet approximately one-third of MSM and one-half of AGYW and female university students in Uganda preferred a 1-week MAP, which they felt would better match their needs since they have sex only intermittently. These respondents felt the short duration would provide the flexibility to discontinue, expressing a preference to not take drugs into their bodies when they did not need them.

"He's never at home. I don't want to use anything when he is absent. I would prefer to use it during that week when he's at home. . .it is less interference with my fertility."–AGYW; Uganda

"I would insist on the 1-week choice. The reason is simple. There is no need taking in too much of anything. Why would I inject myself with drugs for 3 months when I have no clue what may happen in that period?"–AGYW; Uganda

AGYW and FSW who were asked about MAPs as an MPT compared the duration of protection to their experiences with injectable contraceptives. These respondents said that if the MAP had a similar duration to injectables it would feel familiar to something they already knew, which would be helpful. Most of those familiar with injectables used the 3-month injectable.

"It should be like the injectables, which you go to the clinic to get every 2 or 3 months."–AGYW; Mpumalanga, South Africa

**Wear time.** Participants shared preferences regarding three proposed wear times (30 minutes, 1 hour, longer than 1 hour) via FGDs and follow-up interviews after mock use with prototypes. Most respondents across the two countries would be willing to wear a MAP for at least 30 minutes. In South Africa, many FGD participants felt it could be acceptable to wear a MAP for up to 1 hour, while others preferred 30 minutes. Preferences were not specific to user groups or locations. AGYW who preferred 30 minutes gave reasons such as convenience, fit with schedule, and adequate time to deliver the drug. In Uganda, all FGD participants initially stated 30 minutes or less saying "the shorter the better." However, as discussions progressed, participants questioned whether wear time was related to MAP size and duration of

protection. Participants then shifted their perspectives with more than half stating they would wear the MAP longer if it meant a longer duration of protection. One concern raised was that longer wear time might interfere with other daily activities and cause side effects from wearing the MAP longer. After mock use, all participants said they would be willing to wear the MAP for 30 minutes, and nearly half said they would be willing to wear it for 1 hour.

> "Those of us on daily tablets of PrEP would rather wear a big patch for as long as an hour or longer for a longer duration of protection even if the protection is for 1 week, but if it is longer so much the better as an alternative to taking tablets daily."–MSM; Uganda

> "Most people don't like going to the clinic. So, if you were to get it applied at the clinic you don't want to have to wait for a long time. So, 30 minutes is fine."–AGYW; KwaZulu-Natal, South Africa

> "For me I think it should be 30 minutes to be very sure that the medicines have entered; if you can bear taking tablets and injections, why can't I wait for this."–FSW; Uganda

> "As long as my routine duties are not disrupted. There's no problem as long as my daily activities are not affected. If I fail to save the 30 minutes, the rest of my life may be at a greater risk to death. So, there's no problem with 30 minutes."–MSM; Uganda

**Site of application.**   Since these MAPs are in preclinical studies, we do not know if there is a "best" site for MAP application, we asked participants in the FGDs and after mock use where they would prefer to apply the MAP and why. Across both countries, the most preferred sites were upper arm or thigh. Participants noted these sites were discreet, any mark on the skin could be covered with clothing, and the location allowed for self-application. Other sites were discussed, such as forearm, chest, stomach, shoulder, back, and hip, but upper arm or thigh were the most common responses.

> "Because [the thigh] will be easy to press, is not hairy and can be discreet."–AGYW; Mpumalanga, South Africa

> "I chose the arm because I want to hide it. It is easy to put it on the arm."–Male partner; Gauteng, South Africa

## Delivery indicator

The participants in the mock-use exercises evaluated both a "spot crush" and a "slide crush" indicator (as diagrammed in Fig 3) and responded to questions about their experience. In South Africa, most participants across all user groups preferred the "slide-crush" indicator due to fewer steps required to activate the microneedles (four grooves to slide along compared to pressing 16 individual dots). In Uganda, participants preferred the "spot crush" because they felt crushing each spot individually provided some feedback that the MAP had been deployed.

> "I have no problem with the first type [spot crush] because it has many marks that you can press as buttons, which is easy to do. We prefer the one which has buttons to press and ensure the drugs have entered the body, not the one where you press against resistance."–University student, male; Uganda

> "The sliding one because it's simpler; for example, if you have a big [MAP], with the spot one it's going to be too much, it is going to take too long. The sliding one is going to be quicker and simpler."–Health care provider; Mpumalanga, South Africa

Although almost all mock-use exercises participants reported being satisfied with MAP ease of use, nearly half of the participants in both countries were dissatisfied with the level of confidence provided by either the spot crush or slide crush delivery indicator that the drugs would be delivered. Users suggested adding a more pronounced tactile or visual cue, possibly a color change to confirm sufficient pressure had been applied to deliver the dose. Users also wanted a signal when the drug was delivered to alert them when to remove the MAP rather than relying on a watch or a clock to measure time. Comments in the ranking evaluation after mock use indicated that *neither* style provided sufficient confidence. In the product feature ranking exercise, the delivery indicator was rated as "very important" by most participants. The researchers' observations about use errors and difficulties highlighted improvements needed for greater clarity about use and confidence in the delivery indicator.

> "If at all there is a way to signal that the time is done. . .the drug has been administered, that would give me the confidence. Otherwise, someone is left wondering did I apply enough pressure."–University student, female; Uganda

> "Right now, it [the MAP indicator] is all white. There is nothing to determine that the medicine has been administered. So. . .we need a sign. . .to show that it's green or red."–FSW; Uganda

**Instructions for use.**   Participants in the mock-use exercises in both countries had some difficulties with the instructions, but markedly less so in South Africa where the participants had seen a demonstration of how to apply the MAP before mock use. The participants and researchers recommended edits to simplify the text and improve the images for greater understanding of the instructions, especially for end users with limited literacy, who will rely primarily on the images; for example, they recommended colored images rather than line drawings to make the images clearer.

**Packaging.**   Mock-use exercises participants had mixed reactions to the prototype packaging. While some participants appreciated that the prototype packaging was sturdy and kept the MAP clean and safe before use, other participants felt the packaging was too bulky, would not fit into their pockets, or was not appealing or attractive.

> "It is very big, yes imagine you want to go to the pharmacy and buy your MAP, the thing is too big and you can't even keep it."–AGYW; Uganda

> "I am dissatisfied with it. . .one may think you are just holding a biscuit when they see you with this."– MSM; Uganda

**Willingness to make design trade-offs.**   When the researchers asked the participants about needs/preferences for the individual product features, participants quickly understood that some were linked; for example, MAP size with duration of protection, which also influenced choice about where to apply the MAP. Also, while nearly all participants preferred a small MAP size with a short wear time, they also expressed a willingness to make trade-offs. Some respondents said they would be willing to use the medium-size MAP if it offered longer protection than the small one, and even wear it longer.

> "I would pick the big one—the one for 3 months. We would prefer how long it would last so now I wouldn't mind size. No, I prefer the protection now. Rather that. Size is not that important."–Male partner; KwaZulu-Natal, South Africa

"The smallest is the best. But if it comes out and there is only a big option, would still use it. But just have to think about where to put it."–AGYW; KwaZulu-Natal, South Africa

**MAP as an MPT.** The concept of the MAP as an MPT was well received by participants in both countries who viewed it as convenient, easy to use, and timesaving. Male participants were enthusiastic about the concept of an MPT MAP and said this could be a good option for women. Female participants emphasized the need to have both MAP options (HIV PrEP and MPT) so that men could access PrEP via a MAP and so that women who were at risk of HIV but wanted to conceive could have this option as well.

"It's good, it protects you from HIV and having a baby. I want it today. It's easy to use."–AGYW; Mpumalanga, South Africa

"From my perspective, I think it's user friendly. Hormonal contraceptives pills, injections, implants can be difficult to use. . .for example, one needs water and must remember to take [the pills] daily. I personally fear injections and dread the pain and with implants."–AGYW; Uganda

Health care providers in South Africa preferred the MPT option over HIV PrEP alone, noting it would save time and be more convenient to have both HIV prevention and contraception in one product. Providers said this would reduce clinic visits, reduce pill burden, and improve adherence. SRH experts were enthusiastic about the potential for an MPT MAP but indicated that both product options should be available to address the needs of different users.

"I think it is necessary because most people that come for contraceptive are having unprotected sex so this would protect them."–Health care provider; KwaZulu-Natal, South Africa

**Refining MAP vocabulary.** Since MAPs are a technology still under development, we asked the participants for feedback about the language used to describe MAPs. Participants said "patch" is a term they would prefer over "MAP" because "patch" is easier to understand and does not seem like "anything harsh or invasive." Participants, especially end users, noted the concept of needles or microneedles is scary and that researchers should look for a different way to explain how the drug enters the body. Time and again, the word "needles" was seen as a barrier to use. In general, the concept of "prevent or protect" was preferred "because everyone wants protection." Participants also said the terms "feedback indicator" or "delivery indicator" are unfamiliar, too technical and complex, and not widely understood.

**Overall preference for MAPS in Uganda.** In Uganda, participants rated satisfaction with key features after mock use using a 5-point Likert-type scale (ranging from "extremely dissatisfied" to "extremely satisfied"). Results showed that the majority of participants were extremely or very satisfied with the features of the MAP—except for the feedback indicator. Nearly two fifths of the respondents were very dissatisfied with it (Table 3). Participants said the feedback indicator and the images explaining their use were confusing. Nearly all participant categories explained that neither of the two feedback mechanisms gave them sufficient confidence that the MAP would have been correctly administered. Participants recommended this feature be refined to provide more confidence to the user that the drug has been correctly administered. As one of them explained:

"There should be another indicator to show that the medicine has entered; I think it is hard for one to believe that the drugs have entered the body with just this device, to be honest. . .there is no obvious signal to indicate that the drugs have entered."– AGYW; Uganda

Table 3. Uganda: Level of satisfaction with MAP features after mock use (N = 36*).

| | Extremely Satisfied | Very Satisfied | Satisfied | Very dissatisfied | Extremely dissatisfied |
|---|---|---|---|---|---|
| Self-administration | 17 | 8 | 9 | 2 | 0 |
| Packaging | 13 | 8 | 10 | 4 | 1 |
| Ease of disposal | 13 | 11 | 7 | 3 | 2 |
| Ease of use | 12 | 10 | 11 | 1 | 2 |
| Ease of storage | 12 | 9 | 12 | 3 | 0 |
| Visual appeal | 10 | 9 | 11 | 6 | 0 |
| Instructions | 10 | 9 | 8 | 5 | 4 |
| Feedback Indicator | 4 | 5 | 12 | 14 | 1 |

*9 participants from each participant group: FSW, MSM, AGYW, HCP.

## Discussion

This study is the first to explore perceptions of and preferences around MAP technology for the purpose of HIV prevention and as an MPT among user groups at potential risk of HIV infection or both HIV and unintended pregnancy in South Africa and Uganda.

Participants in both countries and across all study groups reported strong interest in and acceptance of the concept of MAPs as a delivery system for HIV PrEP and as an MPT, and positive attitudes about incorporating MAP products into the health care system in the future. As with other assessments of HIV prevention technologies, perceived risk is a key factor influencing potential uptake. FSW and MSM seemed highly motivated to use a MAP because they saw MAP drug delivery as addressing issues with existing HIV prevention options. For example, although long-acting cabotegravir is highly effective for HIV prevention, pain at the injection site and other side effects have dampened acceptability in some populations [61]. And while the dapivirine vaginal ring has been found acceptable in multiple clinical trials [62–65], negative impact on sex and discomfort using the ring during menses have affected acceptability [65]; findings have been more pronounced among women engaged in transactional sex [66]. For AGYW, the potential of an MPT MAP that protects from both HIV and unintended pregnancy seemed to be a strong motivator. Health care providers and other stakeholders such as program managers and policymakers saw value in both the HIV PrEP MAP and the MPT MAP to meet the needs of different user groups.

In our assessment, we found few differences between respondent groups in South Africa and Uganda, despite the slightly different state of the HIV epidemic in the two countries, the status of PrEP rollout, the contraceptive method mix, and slight differences in recruitment of study populations (e.g., South Africa included men as partners of sex workers and female youth activists; Uganda included male and female university students and male youth in the community). This builds confidence that potential user groups across countries might find MAPs acceptable. Users and other stakeholders saw MAPs as having multiple advantages over existing HIV prevention and contraceptive products. MAPs were perceived as discreet, easy to use, reducing daily oral pill burden, potentially painless, and able to be self-administered.

In addition to the overall interest in MAPs, potential users raised concerns, some of which were beyond the scope of this assessment. Potential users wanted to know more about how this new drug delivery system works, what happens to the microneedles after they enter the skin, and any impact if they used MAPs repeatedly over time. They also wanted more confidence that the drug had been delivered and they would be protected. Although we had aimed to use simple language to describe the MAP technology and how it works, users' questions made it clear that these explanations were insufficient. Our findings reinforce the critical

importance of engaging users and other stakeholders early and throughout the product development process.

While almost all potential user groups preferred a MAP with at least a 1-month duration of protection, some AGYW and MSM in Uganda wanted a MAP with a shorter duration (i.e., 1 week). This points to the reality that no one product will meet the needs of all user groups [20,32,67]. While there is strong emphasis among the public health community to develop protection options with a longer duration of protection, some consumers will need or want "on-demand", short-term protection that better fits the realities of their lives. Based on the questions brought forward in this assessment, product developers and researchers should not underestimate the concerns about safety, effectiveness, and side effects of drugs for HIV prevention and contraception. Consumers are concerned about the impact of drugs and do not want to use drugs when they feel they are not at risk.

Although some users preferred the spot crush and others preferred the slide crush indicator, our mock-use exercises showed that the overall indicator design needs refinement to reduce use errors and build confidence that the MAP has been applied correctly and the drug has been delivered. Participants' recommendations to add a more pronounced auditory cue or a color change to indicate that sufficient pressure has been applied or sufficient time has passed would be especially important in communities where users do not commonly have access to a watch or clock to monitor time. Such features, however, could add complexity and cost, contrary to the goal of a low-cost delivery system in countries where the need for HIV prevention and MPTs is greatest. Once again, these findings highlight the importance of incorporating end user research as early as possible to identify features that can be modified during product development for maximum acceptability and usability [68–70].

Mock-use exercise participants in Uganda had more difficulty understanding the instructions for use and experienced more use errors and difficulties with the MAP prototype than participants in South Africa. This may have been because participants in South Africa had been oriented to the MAP technology through their experience in the FGDs and had seen a demonstration of the MAP prior to mock use, or it may also be that more participants in Uganda had lower literacy. Either way, our results indicate that the instructions were insufficient for some user groups without additional coaching or counseling from a provider. Users and research teams from both countries suggested edits to the instructions to simplify the text, especially the technical terminology (e.g., "delivery indicator"), and to improve the images for greater clarity.

Although we asked potential users to provide feedback on individual MAP features, most participants quickly recognized that some product features would be linked (e.g., size and duration of protection). Participants then engaged in a more nuanced discussion around priorities and trade-offs for product features, similar to the way that participants in discrete choice experiments weigh various product characteristics [34,71]. Given users' desire for a small MAP that provides a long duration of protection, the drug loading and consequent MAP size required to deliver a sufficient dose of currently available ARVs such as cabotegravir might be a barrier to acceptance [60]. For this reason, new, more highly potent ARV candidates in the drug development pipeline may be more likely to meet user needs for delivery via MAP.

In both countries, potential users had questions about the drug that would be delivered: what impact the drug would have on their body, and what the safety, side effects, and effectiveness would be. Participants hoped the MAP could deliver an ARV for HIV PrEP and/or progestin for contraception with fewer side effects than existing products. Participants also emphasized the need for both MAP formulations (HIV PrEP and MPT) to address the complexities of individuals' lives.

While there was great interest across users and stakeholders in South Africa and Uganda, who uniformly perceived that the MAP could fit into their health care system and address needs in their countries, there was also a "believability" issue. They felt that a MAP that delivers long-acting protection with no pain was almost too good to be true, asking, "If there is no pain, then how do I know it works?" Future MAP preclinical and clinical studies should assess whether users will see a temporary mark or reaction from the microneedles which could provide a visual cue the drug has been delivered, and the acceptability of such markings.

### Limitations

First, this was an initial assessment to explore the hypothetical acceptability and usability of a prototype MAP. Hypothetical acceptability does not necessarily reflect user acceptability once a product becomes available, however, early end user research can generate critical information that will influence the ultimate product design and increase the likelihood of uptake by users [72–75]. In addition, although we asked potential users to assess MAPs as a delivery system for either HIV PrEP or as an MPT, we did not identify the specific drug(s) to be delivered, and some participants found it difficult to evaluate the drug delivery system separate from the drugs it would deliver, which may limit the utility of the results. Second, while research teams covered the same overall topics to explore acceptability and preferences, the teams worded questions differently, did not ask all the same questions, and incorporated quantitative measurements in different activities. Most significantly, the research team in Uganda included a Likert-style scale to assess satisfaction with product features after mock use. Thus, it was challenging to synthesize findings from the two countries; yet key findings across the two countries were similar. In addition, the results from the mock use exercise in Uganda highlighted that the instructions were insufficient for people who had not been previously oriented to the MAP by participating in FGDs, emphasizing the importance of additional work on refining and testing the instructions. Finally, the findings from our study represent the opinions of a relatively small sample of purposively selected participants and are not generalizable. Despite this limitation, the in-depth feedback from participants could potentially set the stage for a survey incorporating discrete choice experiments to explore trade-offs of MAP features for HIV PrEP and as an MPT among a broader population of end users.

### Conclusion

Across both countries and among all potential user and stakeholder groups, we found a high degree of interest and acceptance of MAP technology. Many potential users and stakeholders saw that MAPs for HIV PrEP could have advantages over existing HIV prevention products and AGYW, FSW, and health care providers were particularly interested in the potential of an MPT MAP. Providers and other key stakeholders felt that an HIV PrEP MAP could be integrated into existing health care systems easily and MPT MAPs could support the aim of integrating FP and HIV services to better serve client needs. Although many participants viewed the MAP as easy to use, feedback from them after a mock use exercise identified areas for further development, including a more obvious indicator of MAP delivery, clearer IFU, and packaging improvements. Product developers used results from this study to refine the MAP design which subsequently was evaluated in a user/stakeholder assessment in Kenya with good results [76]. If a higher-potency ARV is identified that can deliver long-acting protection through a MAP patch size that is acceptable to users, MAPs could improve uptake and acceptability of HIV PrEP for populations at risk for HIV, and an MPT MAP could help address the needs of AGYW and FSW at risk of both HIV and unintended pregnancy in sub-Saharan Africa.

## Acknowledgments

We would like to thank the participants who willingly shared their perspectives and recommendations to improve MAP development, and the research teams who implemented these studies. In addition, we thank Ingela Emblen of PATH for her expert editorial assistance and Annie Rein-Weston, Ben Creelman, Jessica Mistilis, Abra Greene, Priscilla Kwarteng, and Jill Sherman-Konkle of PATH for their technical contributions to the study and review of and support for this manuscript.

This manuscript is dedicated to the memory of David Katuntu, our respected PATH Uganda colleague who passed away in August 2020.

## Author Contributions

**Conceptualization:** Ayesha Ismail, Sarah Magni, Anne Katahoire, Godfrey Siu, Fred Semitala, Peter Kyambadde, Courtney Jarrahian, Maggie Kilbourne-Brook.

**Data curation:** Ayesha Ismail, Sarah Magni, Anne Katahoire, Florence Ayebare.

**Formal analysis:** Ayesha Ismail, Sarah Magni, Anne Katahoire, Florence Ayebare, Godfrey Siu, Fred Semitala, Peter Kyambadde, Barbara Friedland.

**Funding acquisition:** Courtney Jarrahian.

**Investigation:** Ayesha Ismail, Sarah Magni, Anne Katahoire, Florence Ayebare, Godfrey Siu, Fred Semitala.

**Methodology:** Ayesha Ismail, Sarah Magni, Anne Katahoire, Godfrey Siu, Fred Semitala, Peter Kyambadde.

**Project administration:** Ayesha Ismail, Anne Katahoire, Florence Ayebare, Peter Kyambadde.

**Supervision:** Ayesha Ismail, Sarah Magni, Anne Katahoire, Florence Ayebare, Peter Kyambadde, Barbara Friedland, Courtney Jarrahian, Maggie Kilbourne-Brook.

**Validation:** Anne Katahoire, Florence Ayebare, Godfrey Siu, Fred Semitala.

**Writing – original draft:** Ayesha Ismail, Sarah Magni, Anne Katahoire, Florence Ayebare, Maggie Kilbourne-Brook.

**Writing – review & editing:** Ayesha Ismail, Sarah Magni, Anne Katahoire, Florence Ayebare, Barbara Friedland, Courtney Jarrahian, Maggie Kilbourne-Brook.

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
