## [Editor Report · Decision Letter 0]

14 Dec 2022

PONE-D-22-31097Exploring user and stakeholder perspectives from South Africa and Uganda to refine microarray patch development for HIV PrEP delivery and as a multipurpose prevention technologyPLOS ONE

Dear Dr. Kilbourne-Brook,

Thank you for submitting your manuscript to PLOS ONE. After careful consideration, we feel that it has merit but does not fully meet PLOS ONE’s publication criteria as it currently stands. Therefore, we invite you to submit a revised version of the manuscript that addresses the points raised during the review process.

We look forward to receiving your revised manuscript.

Kind regards,

Vincent Setlhare, MD

Academic Editor

PLOS ONE

Journal Requirements:

"These assessments and this manuscript were made possible by the generous support of the American people through the United States Agency for International Development (USAID) through the United States President’s Emergency Plan for AIDS Relief (PEPFAR), under the terms of Cooperative Agreement #AID-OAA-A-17-00015."

"1 of 1 Funder:

-Initial of author who received the award: CJ

-Grant number: AID-OAA-A-17-00015.

-Funder name: US Agency for International Development (USAID) 

-Website: https://www.usaid.gov/

NO. The funder had no role in study design, data collection and analysis, decision to publish, or preparation of the manuscript."

4. We note that you have included the phrase “data not reported here” in your manuscript. Unfortunately, this does not meet our data sharing requirements. PLOS does not permit references to inaccessible data. We require that authors provide all relevant data within the paper, Supporting Information files, or in an acceptable, public repository. Please add a citation to support this phrase or upload the data that corresponds with these findings to a stable repository (such as Figshare or Dryad) and provide and URLs, DOIs, or accession numbers that may be used to access these data. Or, if the data are not a core part of the research being presented in your study, we ask that you remove the phrase that refers to these data.

**Additional Editor Comments:**

Human subjects were used in this study. Please submit IRB approval/s with the manuscript.

---

## [Author Response · Author response to Decision Letter 0]

27 Jan 2023

Review comments and response

We have made minor formatting changes to comply with PLOS ONE’s style requirements. Apologies that we missed these the first time.

2. Thank you for stating the following in the Acknowledgments Section of your manuscript: “These assessments and this manuscript were made possible by the generous support of the American people through the United States Agency for International Development (USAID) through the United States President’s Emergency Plan for AIDS Relief (PEPFAR), under the terms of Cooperative Agreement #AID-OAA-A-17-00015.”

We note that you have provided funding information that is not currently declared in your Funding Statement. However, funding information should not appear in the Acknowledgments section or other areas of your manuscript. We will only publish funding information present in the Funding Statement section of the online submission form. Please remove any funding-related text from the manuscript and let us know how you would like to update your Funding Statement. Currently, your Funding Statement reads as follows: 

“-Initial of author who received the award: CJ

-Grant number: AID-OAA-A-17-00015.

-Funder name: US Agency for International Development (USAID) 

-Website: https://www.usaid.gov/

NO. The funder had no role in study design, data collection and analysis, decision to publish, or preparation of the manuscript.”

Thank you for pointing this out. We have removed the funding language from the manuscript. The Funding Statement in the online submission is correct as stated above.

3. PLOS requires an ORCID iD for the corresponding author in Editorial Manager. 

We have connected the submission with my ORCID ID, which is: 0000-0001-6161-7213 

4. We note that you have included the phrase “data not reported here” in your manuscript. Unfortunately, this does not meet our data sharing requirements. PLOS does not permit references to inaccessible data. We require that authors provide all relevant data within the paper, Supporting Information files, or in an acceptable, public repository. Please add a citation to support this phrase or upload the data that corresponds with these findings to a stable repository (such as Figshare or Dryad) and provide and URLs, DOIs, or accession numbers that may be used to access these data. Or, if the data are not a core part of the research being presented in your study, we ask that you remove the phrase that refers to these data.

Thank you for catching this statement. We have removed the phrase “data not reported here” from the manuscript. Results for all core parts of the study are included in the manuscript.

We checked the references and have made minor revisions as follows:

In the Introduction and Materials and methods sections: 

• We removed a reference that described the “monthly oral pill” that was being developed as an additional HIV PrEP product since the developer halted further development of this ART for HIV PrEP [former reference 13].

• In the same sentence, we added text about new HIV prevention technologies in development and a reference to the MATRIX website [new reference 13]. 

• We added a reference regarding the rationale for multipurpose prevention [new reference 16]. 

• We added references describing development of MAPs for various health indications [new references 33 and 34]

• We updated cross-references to reflect current numbers.

In the References section:

• We added the reference text for the new references noted above—13, 16, 33, 34—and removed other text as needed

• We corrected a link that was not functional in an existing reference [current reference 32]. 

PATH. The PATH Center of Excellence for Microarray Patch Technology: Advancing the Microarray Patch Delivery Technology Platform for High-Priority Global Health Needs. Seattle: PATH; 2019. Available from: https://media.path.org/documents/MAPs-fs-Jan2020.pdf. 

6. Human subjects were used in this study. Please submit IRB approval/s with the manuscript.

Apologies for misunderstanding the prompt. As noted at the end of the Materials and methods section of the manuscript, in South Africa, ethics approval was obtained from the University of Witwatersrand Human Research Ethics Committee (Medical) (Clearance Certificate No. M190721). In Uganda, ethics approved was provided by the Makerere University School of Medicine Research Ethics Committee and the Uganda National Council for Science and Technology (UNCST SS5090 in Uganda. We have submitted the three approval letters as Supporting information (S1, S2, and S3) and corrected the submission coversheet to indicate that these studies included Human Subjects ethics review.

---

## [Decision Letter · Decision Letter 1]

14 Jun 2023

PONE-D-22-31097R1Exploring user and stakeholder perspectives from South Africa and Uganda to refine microarray patch development for HIV PrEP delivery and as a multipurpose prevention technologyPLOS ONE

Dear Dr. Kilbourne-Brook,

Thank you for submitting your manuscript to PLOS ONE. After careful consideration, we feel that it has merit but does not fully meet PLOS ONE’s publication criteria as it currently stands. Therefore, we invite you to submit a revised version of the manuscript that addresses the points raised during the review process.

ACADEMIC EDITOR:I am in agreement with the two reviewers that this manuscript has merit. Please attend to their suggested edits and additions, particularly Reviewer 2's request for expansion in the Discussion. I do not have additional suggestions.==============================

We look forward to receiving your revised manuscript.

Kind regards,

Sara Jewett Nieuwoudt, Ph.D, MPH

Academic Editor

PLOS ONE

Journal Requirements:

Reviewers' comments:

Reviewer's Responses to Questions

**Comments to the Author**

1. If the authors have adequately addressed your comments raised in a previous round of review and you feel that this manuscript is now acceptable for publication, you may indicate that here to bypass the “Comments to the Author” section, enter your conflict of interest statement in the “Confidential to Editor” section, and submit your "Accept" recommendation.

Reviewer #1: (No Response)

Reviewer #2: (No Response)

2. Is the manuscript technically sound, and do the data support the conclusions?

Reviewer #1: Yes

Reviewer #2: Yes

3. Has the statistical analysis been performed appropriately and rigorously? 

Reviewer #1: Yes

Reviewer #2: I Don't Know

4. Have the authors made all data underlying the findings in their manuscript fully available?

Reviewer #1: Yes

Reviewer #2: Yes

5. Is the manuscript presented in an intelligible fashion and written in standard English?

Reviewer #1: Yes

Reviewer #2: Yes

6. Review Comments to the Author

Reviewer #1: Manuscript is well written; the sections are clearly elaborated. The objectives are stated within the introduction and methods setion is technically sound in terms of sample sizes, procedures and data analysis.

Statistical analysis differed by site.

Generally, the manuscript is well organised and written out.

Abstract:

Line 25-26: It is not clear to which group the male partners were. "Participants included adolescent girls and young women, female sex workers and male partners of these women as well as men who have sex with men."

Methods:

Line 123: Review the epidemological research design. "We used a cross sectional qualitative research design or approach ..."

Line 196-197, Table 2 on page 12: Inconsistences in the listed participant groups as stated in abstract and methods sections. AGYW (COmmunity), AGYW (Univesity)

Line 26 and Table 12 on page 12: What was the criterion of including the men participants?

Results:

Page 11: Demographic information to introduce the participants is sometimes helpful. Insert that paragraph.

Page 11 and Table 2: Table 2 seems to fit better in data management and analysis section

Reviewer #2: General comment

This is an exciting and informative manuscript. It addresses an important topic significant for HIV prevention research and creating HIV prevention options. It focuses on the acceptability of micro-array patches (MAP) to prevent HIV and MAPs as multi-purpose technologies, including family planning. This manuscript contributes significantly to the literature and development of a new HIV prevention method - MAP. Here are my comments.

Abstract

• Add a line about data analysis in the abstract.

• Line 35, acceptable?

Introduction

• Line 75, spell out and define MPTs.

Methods

• Line 175- specify the local languages.

• Describe the setting of mock exercises, IDIs, and FGDs where were they done? The FGDs and mock exercises were made up of how many people? How were they stratified? How long did they (IDIs, FGDs, and Mock use sessions) take?

• The authors mentioned that a survey (Likert Scale) was done in Uganda to assess mock-use exercises. However, there is no information on how that data was analyzed. Can we actually say this was a purely qualitative study if it included quantitative approaches like the Likert scale?

Results

• Line 261- spell out/ define IFU.

• Line 272- what kind of challenges do MAPs address? Provide examples.

• Were they any differences in perceptions and preferences between groups and countries

Discussion

This section is missing reference to literature on current studies on other effective HIV prevention methods, e.g., vaginal ring, CAB-LA, etc. For instance, the authors do mention that it addresses issues seen in other HIV prevention methods, but it’s unclear what issues and in what way MAPs address those issues.

7. PLOS authors have the option to publish the peer review history of their article (what does this mean?). If published, this will include your full peer review and any attached files.

Reviewer #1: No

Reviewer #2: No

---

## [Author Response · Author response to Decision Letter 1]

31 Jul 2023

RESPONSE TO REVIEWERS

REVIEWER #1: 

ABSTRACT

Line 25-26: It is not clear to which group the male partners were. "Participants included adolescent girls and young women, female sex workers and male partners of these women as well as men who have sex with men."

RESPONSE: Thank you for this comment. We have revised the abstract per your suggestion.

METHODS

Line 123: Review the epidemiological research design. "We used a cross sectional qualitative research design or approach..."

RESPONSE: We have revised this sentence per your suggestion to tighten the description of the research design, and to respond to a question from Reviewer #2. Thank you. 

Line 196-197, Table 2 on page 12: Inconsistences in the listed participant groups as stated in abstract and methods sections. AGYW (Community), AGYW (University)

RESPONSE: Thank you for pointing out this discrepancy. We revised the abstract to clarify these groups and make the text consistent. 

Line 26 and Table 2 on page 12: What was the criterion of including the men participants?

RESPONSE: Participants recruited for this assessment were aged 18–40 years, and included young women and men from key populations and vulnerable groups at risk of HIV and/or unintended pregnancy. We recruited men both for their potential role as “end-users” of an HIV PrEP MAP and also as male partners who influence women’s choice and ability to use HIV prevention strategies and contraception. We added a Table 2 footnote to clarify. 

RESULTS

Page 11: Demographic information to introduce the participants is sometimes helpful. Insert that paragraph.

RESPONSE: We added detail at the beginning of the Results section to clarify the demographic profile of study participants. 

Page 11 and Table 2: Table 2 seems to fit better in data management and analysis section.

RESPONSE: Thank you for this comment. We believe Table 2 belongs in the Results section because it provides an overview of the research activities that were completed, by activity and by site. The table title was confusing, so we have revised the table title to better reflect the content. 

REVIEWER #2:

ABSTRACT

Add a line about data analysis in the abstract.

RESPONSE: We added sentence to the abstract to address this comment.

Line 35: acceptable?

RESPONSE: Thank you for this suggestion that the word “acceptable” might be a better choice than “desirable”. The response among the potential user groups and stakeholders was stronger than just acceptable. Self- administration is perceived as a valued characteristic of a MAP product, so we have used this word. 

INTRODUCTION

Line 75: spell out and define MPTs.

RESPONSE: Thank you for catching this. We’ve spelled out multipurpose prevention technologies (MPT), as this is the first use of this abbreviation in the manuscript.

METHODS

Line 175: specify the local languages.

Setting of activities: Describe the settings of mock exercises, IDIs, and FGDs where were they done? The FGDs and mock exercises were made up of how many people? How were they stratified? How long did they (IDIs, FGDs, and Mock use sessions) take?

RESPONSE: Thank you for these comments/recommended edits. We revised the text to address these points.

Likert-style scale: The authors mentioned that a survey (Likert Scale) was done in Uganda to assess mock-use exercises. However, there is no information on how that data was analyzed. Can we actually say this was a purely qualitative study if it included quantitative approaches like the Likert scale?

RESPONSE: We used the Likert-style scale in Uganda to help participants focus their perspectives about “satisfaction” with MAP features. We used this tool to facilitate the debrief interview regarding features and whether they met participants’ needs, and to highlight priority areas for future refinements. For clarity, we revised the Study design text. Also, we added a statement in Data Management and Analysis section to clarify.

RESULTS

Line 261: spell out/define IFU.

RESPONSE: We have defined IFU in this line, per your request.

Line 272: what kind of challenges do MAPs address? Provide examples.

RESPONSE: Thank you for this question. We tried to address this in the Introduction section where we describe the potential value of MAPs for drug delivery (see paragraph 5 of the Introduction section), but we have added specific examples in these perceived benefits relative to existing HIV prevention and contraceptive methods in the Results section as well, per your request.

Were there any differences in perceptions and preferences between groups and countries?

RESPONSE: Thank you for this question. We did not find significant differences between countries or user groups. We saw strong similarities between user groups stated needs and preferences for HIV prevention and MPT products. We did find some nuances between groups, for example FSW and MSM, who already were aware of their risk of HIV and were more motivated to use HIV prevention, were more willing to accept MAP—even if it was larger than they preferred. They were more willing to accept a larger sized MAP especially if it could provide longer term protection. AGWY were not as willing to accept a large sized MAP. At this early stage of product development, when the active pharmaceutical ingredient is not yet identified, we felt the major findings of similarities were more relevant to report than the nuanced findings. Our interpretation of these findings is already included in the Discussion section, paragraphs 2–7 (revised lines 628–691).

DISCUSSION

This section is missing reference to literature on current studies on other effective HIV prevention methods, e.g., vaginal ring, CAB-LA, etc. For instance, the authors do mention that it addresses issues seen in other HIV prevention methods, but it’s unclear what issues and in what way MAPs address those issues.

RESPONSE: We have added detail and citations to the Introduction and Discussions section to address the current state of oral PrEP introduction and other prevention options, such as dapivirine vaginal ring and long-acting injectable cabotegravir. Thank you.

---

## [Editor Report · Decision Letter 2]

11 Aug 2023

Exploring user and stakeholder perspectives from South Africa and Uganda to refine microarray patch development for HIV PrEP delivery and as a multipurpose prevention technology

PONE-D-22-31097R2

Dear Dr. Kilbourne-Brook,

We’re pleased to inform you that your manuscript has been judged scientifically suitable for publication and will be formally accepted for publication once it meets all outstanding technical requirements.

Kind regards,

Sara Jewett Nieuwoudt, Ph.D, MPH

Academic Editor

PLOS ONE

Editor Comments:

Thank you for taking the time to respond to the reviewer comments comprehensively and in detail. Your letter of responses was very helpful in the latest round of reviews.

---

## [Editor Report · Acceptance letter]

23 Aug 2023

PONE-D-22-31097R2 

Exploring user and stakeholder perspectives from South Africa and Uganda to refine microarray patch development for HIV PrEP delivery and as a multipurpose prevention technology 

Dear Dr. Kilbourne-Brook:

I'm pleased to inform you that your manuscript has been deemed suitable for publication in PLOS ONE. Congratulations! Your manuscript is now with our production department. 

Kind regards, 

on behalf of

Dr. Sara Jewett Nieuwoudt 

Academic Editor

PLOS ONE